# The Role of Nondiabetic Hyperglycemia in Critically Ill Patients with Acute Ischemic Stroke

**DOI:** 10.3390/jcm11175116

**Published:** 2022-08-30

**Authors:** Hung-Sheng Shih, Wei-Sheng Wang, Li-Yu Yang, Shu-Hao Chang, Po-Huang Chen, Hong-Jie Jhou

**Affiliations:** 1Department of Biomechatronic Engineering, National Taiwan University, Taipei 10617, Taiwan; 2Department of Neurology, Changhua Christian Hospital, Changhua 500, Taiwan; 3School of Medicine, Kaohsiung Medical University, Kaohsiung 80756, Taiwan; 4Department of Computer Science and Information Science, National Formosa University, Yunlin 63201, Taiwan; 5Department of Internal Medicine, Tri-Service General Hospital, National Defense Medical Center, Taipei 114, Taiwan

**Keywords:** ischemic stroke, hyperglycemia, non-diabetes, MIMIC-IV, propensity score matching

## Abstract

In this study, we aim to elucidate the association between nondiabetic hyperglycemia and the short-term prognosis of critically ill patients with acute ischemic stroke. We extracted data using the Medical Information Mart for Intensive Care IV from 2008 to 2019. The primary outcomes were set as intensive care units (ICU) and in-hospital mortality. We developed a Cox proportional hazards model to determine the nonlinear association between serum glucose levels and primary outcomes. Of the 1086 patients included, 236 patients had hyperglycemia. Patients with hyperglycemia were associated with higher ages, female gender, higher Charlson Comorbidity Index scores, and higher Acute Physiology Score III scores. After propensity score matching, 222 pairs remained. The hyperglycemia group had a significantly higher ICU mortality (17.6% vs. 10.8%; *p* = 0.041). Meanwhile, no significant differences in ICU length of stay (5.2 vs. 5.2; *p* = 0.910), in-hospital mortality (26.6% vs. 18.9%, *p* = 0.054), and hospital length of stay (10.0 vs. 9.1; *p* = 0.404) were observed between the two groups. The Kaplan–Meier curves for ICU and in-hospital survival before matching suggested significant differences; however, after matching, they failed to prove any disparity. Non-diabetic patients with acute ischemic stroke have poor clinical characteristic while encountering hyperglycemic events; therefore, careful monitoring in the acute phase is still required.

## 1. Introduction

Worldwide, stroke remains the second-leading cause of death and the third-leading cause of disability [1]. There are 12.2 million new cases of stroke per year, and the number of individuals living with stroke has almost doubled over the past three decades [2]. Ischemic stroke accounts for approximately 90% of all stroke cases [3]. In the United States, there are more than 700,000 new cases of acute ischemic stroke (AIS) and recurrent ischemic stroke annually [4]. However, although the mortality rate after AIS is relatively low in a real-world experience (less than 6%), it could lead to a two-fold increase among critically ill AIS patients [5,6].

Poststroke hyperglycemia (PSH) is a common complication that is associated with poor prognosis. Alterations in glucose metabolism due to fluctuations in peripheral insulin resistance and endocrine interactions may result in stress hyperglycemia [7]. Persistent hyperglycemia alters mitochondrial function and increases free radicals, which can aggravate brain ischemic cascades and further compromise oxidative status. This is presumably because of the increased hyperosmolarity, anaerobic metabolism, and focal toxicity associated with ischemic vascular disease [8,9]. Moreover, animal and human studies have also pointed to the association between stress hyperglycemia and elevated serum lactate levels, which result in tissue acidosis and brain hypoxia [10]. Therefore, PSH has been recognized as a prognostic factor that is associated with increased mortality and morbidity.

A recently published review has revealed that more than one-third of critically ill patients were hyperglycemic [11]. In other observational studies, the prevalence of hyperglycemia ranged from 32% to 38% and even reached 16% in patients without diabetes [12]. According to the American Heart Association and American Stroke Association guidelines, treating hyperglycemia to achieve serum glucose levels ranging between 140 and 180 mg/dL (7.8–10.0 mmol/L) and closely monitoring patients to prevent severe hypoglycemia are recommended; however, the level of evidence is based on limited supporting data [13]. In addition, the definition of hyperglycemia in patients without diabetes is inconsistent among studies. An arbitrary cut-point value may result in inaccurate risk estimation. We, therefore, aim to study the association between hyperglycemia and non-diabetic patients suffering from AIS and to elucidate the relationship between those characteristics and subsequent clinical outcomes, hoping to provide some recommendations for physicians in daily clinical practice.

## 2. Materials and Methods

### 2.1. Data Source

A single-center and retrospective matched-cohort study was conducted based on the Medical Information Mart for Intensive Care (MIMIC)-IV database (version 1.0) [14]. The MIMIC-IV database is an updated version of the MIMIC-III database, which provided deidentified critical care data, according to the Health Insurance Portability and Accountability Act Safe Harbor provision, for over 40,000 patients admitted to the Emergency Department or intensive care units (ICUs) of the Beth Israel Deaconess Medical Center (BIDMC) between 2008 and 2019. Under the pre-existing Institutional Review Board (IRB) approval from the Massachusetts Institute of Technology and BIDMC [15], several improvements, including structure simplifying, expansion of new data elements, and improvement of usability, have been made. One author, Hong-Jie Jhou, achieved access to the database to extract data under the certification of the Collaborative Institutional Training Initiative examination (number: 39050603). The study protocol was approved by the IRB of Changhua Christian Hospital (IRB No. 211106).

### 2.2. Study Populations and Variable Extraction

We extracted data of patients from the MIMIC-IV database, focusing on those aged between 18 and 89 years, who were at one time admitted to the ICU under the diagnosis of ischemic stroke, defined as the International Classification of Disease-10 codes of I63, I65, and I66 or the International Classification of Disease-9 codes of 433, 434, 436, 437.0, and 437.1. Moreover, patients who received reperfusion therapy, intravenous injection of recombinant tissue plasminogen activator (rtPA), or mechanical thrombectomy were enrolled. We excluded 536 patients with diabetes and 40 patients without blood sugar data. The final sample included 1086 patients (Figure 1).

The following patient characteristics were collected: (1) demographic characteristics, including age, gender, and race; (2) comorbidities, including hypertension, hyperlipidemia, congestive heart failure, coronary artery disease, peripheral vascular disease, liver disease, peptic ulcer disease, chronic obstructive pulmonary disease, renal disease, malignancy, and rheumatoid disease; (3) the maximum value of laboratory data and vital signs within the first 24 h of ICU stay, including heart rate, respiratory rate, mean arterial pressure, body temperature, leukocyte count, platelet count, hemoglobin, urea nitrogen, creatinine, sodium, potassium, and bilirubin; (4) clinical management, including the use of sedatives, vasopressors, anti-platelet drugs (e.g., aspirin, clopidogrel, ticagrelor, prasugrel, ticlopidine, cilostazol, and dipyridamole), anti-coagulants (e.g., warfarin, dabigatran, rivaroxaban, edoxaban, and apixaban), reperfusion therapy (e.g., rtPA and mechanical thrombectomy), tracheostomy, or percutaneous endoscopic gastrostomy/jejunostomy tube placement; (5) severity and comorbidity scoring systems, including the Charlson Comorbidity Index (CCI) [16,17], and Acute Physiology Score III (APS III) [18]. The sample along with the characteristic variables, aforementioned predictors, and outcome measures were extracted via a Structured Query Language script.

### 2.3. Outcome Measures

The primary outcomes were ICU mortality and in-hospital mortality defined as any cause of death after stroke. The secondary outcomes included ICU length of stay, hospital length of stay, the incidence of intracerebral hemorrhage, the need of percutaneous endoscopic gastrostomy/jejunostomy tube placement, and the need of tracheostomy. The table named “patients” and “admissions” in the MIMIC-IV database provided the survival information and the length of hospital stay [15]. We adhered to the STROBE guidelines for the reporting of this cohort study [19].

### 2.4. Statistical Analysis

Demographics and comorbidity were provided as frequencies and proportions for categorical data and means ± standard deviations for continuous variables, as appropriate. We used the unpaired *t*-test or Mann–Whitney U-test for continuous variables and the chi-square test and Fisher’s exact test for categorical variables. 

The restricted cubic splines were conducted to detect the possible nonlinear dependance of the association between hyperglycemia and ICU and in-hospital mortality among patients with ischemic stroke. There was a sigmoid curve between PSH and in-hospital mortality (*p* value for nonlinear dependance < 0.001) (Figure 2). Then, the patients were categorized according to serum glucose level—below 140 mg/dL (non-hyperglycemia) and above 140 mg/dL (hyperglycemia)—based on the trend implied by our model.

We conducted propensity score matching (PSM) analysis to reduce selection bias. The propensity scores involved the four following variables: age, gender, APS III, and CCI. The patients’ propensity scores were calculated using the logistic regression model for the whole cohort [20]. Matching was developed using the Greedy 5-to-1 Digit-Matching algorithm between the hyperglycemia and non-hyperglycemia groups [21]. After PSM, 222 matched pairs were created. For the comparison of the matched cohort, the paired *t*-test or Wilcoxon signed-rank test was used for continuous variables and McNemar’s test was used for categorical variables. The distribution of propensity scores was provided to evaluate the effectiveness of the PSM. All outcomes were compared based on the matched data.

For time-to-event outcomes, the primary outcomes were estimated using the Kaplan–Meier method and compared using the log-rank test. Univariate and multivariate Cox hazards model analyses were performed to identify the association between hyperglycemia and primary outcomes, and the results were expressed as hazard ratios (HRs) with 95% confidence intervals (CIs). All comparisons were planned, the tests were two-sided, and *p*-values of less than 0.05 were used to indicate statistical significance. All analyses were conducted using R statistical software (version 4.0.3; R Foundation for Statistical Computing, Vienna, Austria).

## 3. Results

### 3.1. Pre-PSM Characteristics and Outcome Measurement

Of the 257,366 medical records reviewed, 50,048 patients were at one time admitted to the ICU. In total, 1662 patients fulfilled our inclusion criteria, of whom 536 were excluded due to a previous diagnosis of diabetes mellitus and 40 lacked blood sugar data. Ultimately, 1086 patients were included in this study, 236 patients were further assorted into the hyperglycemia group, and 850 patients were classified into the non-hyperglycemia group (Figure 1). The basic demographic characteristics of the patients are shown in Table 1.

### 3.2. Post-PSM Characteristics and Outcome Measurement

PSM based on age, gender, APS III, and CCI resulted in the creation of 222 matched pairs. Both the hyperglycemia and non-hyperglycemia groups were well-balanced with the four covariates (Figure 3). After PSM, the hyperglycemia group still showed significantly higher ICU mortality (17.6% vs. 10.8%; *p* = 0.041). Meanwhile, in terms of ICU length of stay (5.2 vs. 5.2 days; *p* = 0.910), in-hospital mortality (26.6% vs. 18.9%; *p* = 0.054), and hospital length of stay (10.0 vs. 9.1 days; *p* = 0.404), no significant differences were observed between the two groups. Regarding the secondary outcomes, the incidence of intracranial hemorrhage (12.2% vs. 13.1%; *p* = 0.775), the need for tracheostomy (7.2% vs. 4.5%; *p* = 0.225), and the need for percutaneous endoscopic gastrostomy/jejunostomy tube placement (17.6% vs. 16.2%; *p* = 0.704) did not significantly differ between the two groups (Table 1).

### 3.3. Kaplan–Meier Survival Curves and Univariate Cox Regression Analysis of Primary Outcomes

To evaluate survival, the patients were followed up until discharge, and the longest hospital length of stay was 88 days. The Kaplan–Meier curves for ICU and in-hospital survival before PSM between the hyperglycemia and non-hyperglycemia groups are shown in Figure 4A,B, which suggested significant differences. Nevertheless, the Kaplan–Meier curves for ICU and in-hospital survival after PSM (Figure 4C,D) failed to show a significant difference between the hyperglycemia and non-hyperglycemia groups.

The results of the univariate Cox regression analysis demonstrated that before PSM, hyperglycemia was related to higher ICU mortality (HR, 2.55; 95% CI, 1.67–3.89) and in-hospital mortality (HR, 2.51; 95% CI, 1.79–3.52); yet, after PSM, ICU mortality (HR, 1.54; 95% CI, 0.92–2.57), and in-hospital mortality (HR, 1.33; 95% CI, 0.90–1.98) did not significantly differ between the two groups (Table 2).

## 4. Discussion

This study revealed that the clinical characteristics of the patients without diabetes with stroke and hyperglycemia and those without hyperglycemia were different. Those with hyperglycemia tended to be associated with higher ages, female gender, higher CCI scores, and higher APS III scores. Because of these differences, patients with hyperglycemia had higher risks of ICU and in-hospital mortality and longer length of stay in the ICU and hospital than those without hyperglycemia. However, after PSM, no significant differences were observed in most primary and secondary outcomes between the two groups, except the ICU mortality in the matched cohort.

The underlying mechanisms linking hyperglycemia to the risk of complications are probably multifactorial and complicated. These events may be due to stress hyperglycemia, which is caused by insulin resistance, the interplay among hormones, and regulatory cytokines. Studies have implicated that hyperglycemia in inflammatory response is associated with poor outcomes [22]. Increased mortality and elevated glucose levels have been associated with impaired brain–blood barrier integrity, which allows the influx of inflammatory cells and blood solutes into the brain, leading to adverse stroke outcomes such as neuronal death and brain edema [23,24]. Additionally, hyperglycemia accelerates damage to the brain tissue surrounding the damaged neurons [25]. Elevated levels of reactive oxygen species are produced by nicotinamide adenine dinucleotide phosphate oxidase and protein kinase C, resulting in possible neuronal damage and reduced reperfusion [26]. Diffusion- and perfusion-weighted magnetic resonance imaging were used to prove the aforementioned findings. Acute hyperglycemia upon admission is correlated with reduced penumbra, a greater final infarction size, and more dependent functional outcomes [8]. Thus, multiple complex molecular mechanisms and pathological changes may contribute to the poor prognosis of stress hyperglycemia after ischemic stroke.

Hyperglycemia is a common complication after ischemic stroke [11]. A recent study has revealed that the prevalence of hyperglycemia is 22% in non-diabetic AIS patients [27,28]. Similar trends were reproduced in our preliminary analysis (21.8%). According to the clinical recommendation from the American Diabetes Association [29], hyperglycemia is defined as serum glucose levels of more than 140 mg/dL (7.8 mmol/L) among hospitalized patients, and the target range is 140–180 mg/dL (7.8–10.0 mmol/L). A similar perspective was also supported by the NICE-SUGAR trial [30]. However, for PSH, the definition varies widely with the cutoff points ranging from 110 mg/dL (6.1 mmol/L) to 180 mg/dL (10.0 mmol/L) [31,32,33,34], which is not a precise guidance. In our analysis, the regression model demonstrated sigmoid curves between the glycemic status and mortality; in addition, the HR of mortality was approximately two-fold higher when the glucose levels reached 140 mg/dL. Consequently, it is reasonable to regard the maximum value of serum glucose in the first 24 h after AIS as a prognostic factor. Furthermore, continuous monitoring to prevent patients’ serum glucose level from falling into the bilateral plateaus of our sigmoid curves should be kept in mind.

In the Stroke Hyperglycemia Insulin Network Effort (SHINE) randomized clinical trial, among patients with AIS with hyperglycemia, neither intensive nor standard glycemic control revealed a significant difference in functional outcomes at 3 months [35]. Meanwhile, another study has proposed that strict glycemic control using insulin improved the National Institutes of Health Stroke Scale (NIHSS) after 30 days [36]. In the UK Glucose Insulin in Stroke Trial (GIST-UK), insulin-based regimens indeed lowered the mean plasma glucose level; however, the 90-day mortality did not show a significant reduction [37]. To sum up, the clinical benefit of intense glucose level control remains ambiguous. Yet, hypoglycemia events were observed more frequently in the strictly controlled group, which is highly correlated with subsequent mortality and morbidity [38]. To some degree, our results also worked in concert with SHINE and GIST-UK. After excluding the clinical heterogeneity, hyperglycemia was not independent of the short-term prognosis in non-diabetic AIS patients. As for diabetic AIS patients, hypoglycemia events are frequent due to defective glucose counter regulation in advanced diabetes mellitus. This evidence makes physicians more distressed in determining whether to treat PSH or not. In this study, we realized that the HR of mortality was low when the glucose level was below 140 mg/dL; moreover, the HR showed a mild increase when the curves approached the left side. Similar viewpoints were confirmed in the latest subgroup analysis of SHINE [38].

In a multicenter, prospective cohort study by the Korean Stroke Cohort for Functioning and Rehabilitation, Yoon et al. highlighted that PSH also affects long-term functional outcomes [27]. A similar current was also noted in other cardiovascular diseases, such as acute coronary syndrome [39,40] and small-vessel disease, especially non-fatal lacunar stroke [41,42], which showed long-term protective effects and even regenerative potential with strict glycemic control. Because of the nature of the MIMIC-IV database, we chose intracranial hemorrhage, tracheostomy, and percutaneous endoscopic gastrostomy/jejunostomy tube placement as the indicators of functional outcomes. Among them, intracranial hemorrhage plays a key role, especially because it significantly affects a patient’s disability and quality of life [43,44]. Even though this study failed to show significant differences after PSM, we should always consider that elevated glucose levels put patients at risk of hemorrhagic transformation, particularly those receiving reperfusion therapy.

The advantages of this study were as follows: this was a broad-scale study based on real-world patient data and this study explored the association between hyperglycemia and survival status among patients with AIS. Nevertheless, the results should be clarified in the context of the several limitations. First, selection bias existed because the clinical information was obtained from a single institution. Second, patients with ischemic stroke were defined according to the first sequence diagnosis code in this retrospective study; however, the diagnostic accuracy remained unknown, and misclassifications may result in false associations. Third, retrospective studies may have baseline differences; therefore, we adjusted as many potential confounders as possible to achieve an appropriate balance using PSM. Fourth, given the limitations of the MIMIC-IV database, some important data were lacking, including the severity scale (e.g., NIHSS), the subtypes of ischemic stroke, cardiac parameters, routine biochemistry exams for stroke risk assessment (e.g., HbA1c value), the management of hyperglycemia, and the cause of mortality. No long-term follow-up data were provided; thus, we cannot obtain the modified Rankin Scale score at 3 months. Finally, because of the nature of observational databases, further randomized clinical trials are needed to validate our findings.

## 5. Conclusions

This retrospective study revealed that non-diabetic AIS patients with hyperglycemia had inferior clinical characteristics compared with those without hyperglycemia. However, after eliminating the clinical heterogeneity, the short-term prognosis was non-significantly different. Overall, sigmoid curves showed that the HR was approximately two-fold higher when the serum glucose level reached 140 mg/dL. It would require early, continuous, and careful monitoring in patients without diabetes with ischemic stroke to avoid critical deterioration. Yet, it is still controversial whether there is a need for strict management of hyperglycemic events for non-diabetic AIS patients.

## Figures and Tables

**Figure 1 jcm-11-05116-f001:**
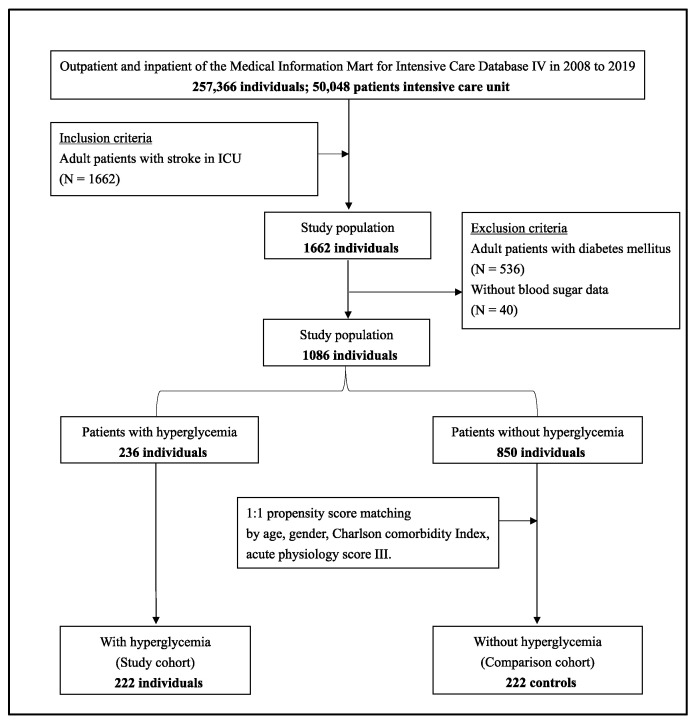
Flow diagram of data extraction from the Medical Information Mart for Intensive Care-IV database and further grouping layout.

**Figure 2 jcm-11-05116-f002:**
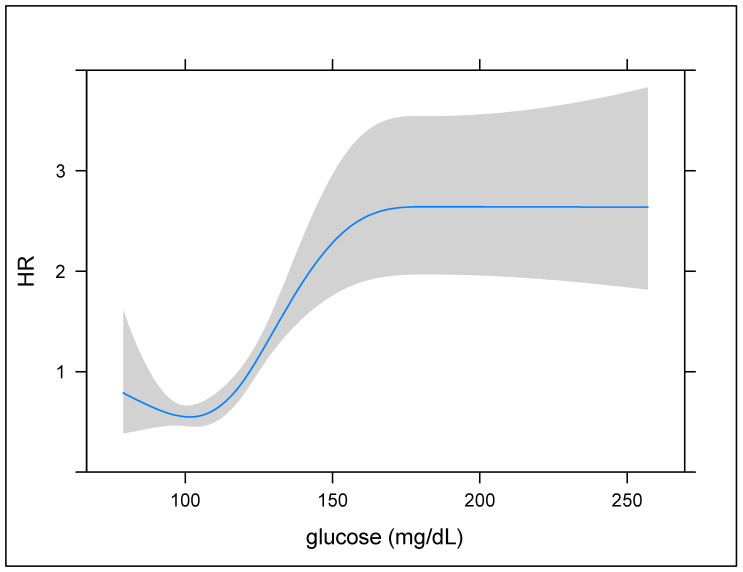
The restricted cubic splines demonstrating the nonlinear relationship between hyperglycemia and in-hospital mortality. The shaded areas around the curves describe the 95% confidence interval.

**Figure 3 jcm-11-05116-f003:**
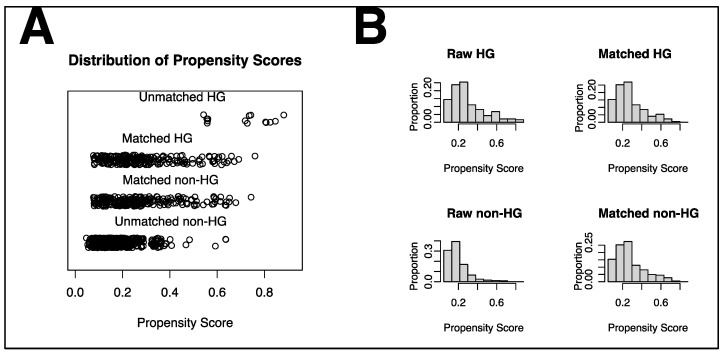
Distribution of propensity scores. (**A**) Jittered plot presenting matched and unmatched subjects, and the distribution of propensity score values. (**B**) Histograms demonstrating the density of propensity score distribution in the hyperglycemia and non-hyperglycemia groups before and after matching. HG, hyperglycemia.

**Figure 4 jcm-11-05116-f004:**
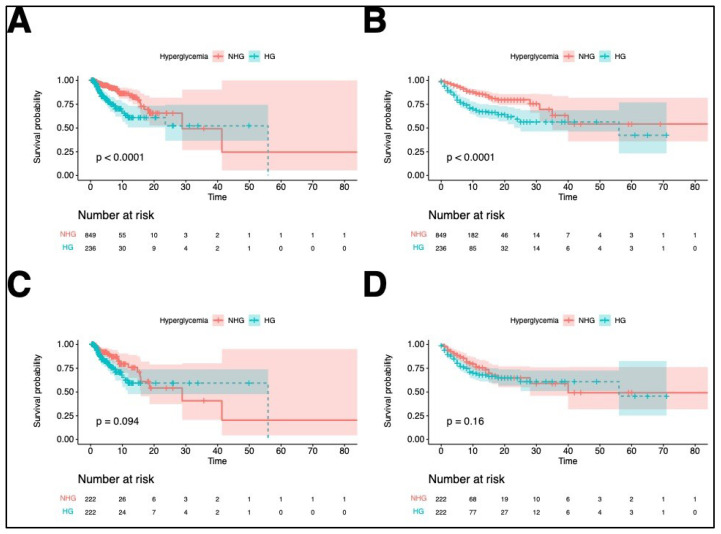
Kaplan–Meier survival curves until intensive care unit (ICU) discharge and hospital discharge. (**A**) ICU mortality before propensity score matching (PSM); (**B**) in-hospital mortality before PSM; (**C**) ICU mortality after PSM; (**D**) in-hospital mortality after PSM. The colored areas describe the standard deviation. ICU, intensive care unit.

**Table 1 jcm-11-05116-t001:** Characteristics of the study patients.

	All Patients			Propensity-Matched Pairs		
Characteristics	HG Group (n = 236)	Non-HG Group (n = 849)	*p* Value	HG Group (n = 222)	Non-HG Group (n = 222)	*p* Value
Age (years)	71.3 ± 13.6	67.2 ± 15.2	<0.001	71.2 ± 13.6	72.2 ± 14.3	0.440
Gender, n			0.011			0.703
Male	106 (44.9%)	461 (54.3%)		101 (45.5%)	105 (47.3%)	
Female	130 (55.1%)	388 (45.7%)		121 (54.5%)	117 (52.7%)	
Race, n			0.239			0.556 #
White	151 (64.0%)	558 (65.7%)		143 (64.4%)	131 (59.0%)	
Black	22 (9.3%)	61 (7.2%)		20 (9.0%)	21 (9.5%)	
Asian	8 (3.4%)	13(1.5%)		7 (3.2%)	4 (1.8%)	
Other	55 (23.3%)	217 (25.6%)		52 (23.4%)	66 (29.7%)	
MAP (mmHg)	117.2 ± 21.0	115.4 ± 19.9	0.231	117.0 ± 21.2	115.4 ± 20.7	0.441
Temperature (°C)	37.5 ± 0.6	37.3 ± 0.5	<0.001	37.5 ± 0.6	37.4 ± 0.6	0.035
Heart rate (beats/min)	105.3 ± 21.0	94.0 ± 19.2	<0.001	104.3 ± 20.8	97.0 ± 23.0	<0.001
Respiratory rate (breaths/min)	27.7 ± 5.7	26.6 ± 5.8	0.010	27.4 ± 5.5	27.6 ± 6.4	0.792
Comorbidities, n						
CCI	6.90 ± 2.50	6.07 ± 2.37	<0.001	6.90 ± 2.50	6.83 ± 2.32	0.783
Hypertension	170 (72.0%)	582 (68.6%)	0.305	161 (72.5%)	165 (74.3%)	0.667
Hyperlipidemia	54 (22.9%)	160 (18.8%)	0.168	50 (22.5%)	41 (18.5%)	0.290
Coronary artery disease	32 (13.6%)	78 (9.2%)	0.049	31 (14.0%)	26 (11.7%)	0.478
Congestive heart failure	46 (19.5%)	128 (15.1%)	0.102	43 (19.4%)	44 (19.8%)	0.905
PVD	24 (10.2%)	103 (12.1%)	0.407	24 (10.8%)	24 (10.8%)	1.000
COPD	36 (15.3%)	125 (14.7%)	0.839	34 (15.3%)	39 (17.6%)	0.552
Liver disease						
Mild	5 (2.1%)	21 (2.5%)	0.753	5 (2.3%)	7 (3.2%)	0.558
Moderate to severe	1 (0.4%)	4 (0.5%)	1.000 #	1 (0.5%)	2 (0.9%)	1.000 #
Peptic ulcer disease	7 (3.0%)	4 (0.5%)	0.003 #	6 (2.7%)	1 (0.5%)	0.122 #
Renal disease	22 (9.3%)	87 (10.2%)	0.676	19 (8.6%)	32 (14.4%)	0.053
Rheumatoid disease	7 (3.0%)	19 (2.2%)	0.518	7 (3.2%)	5 (2.3%)	0.558
Malignancy	30 (12.7%)	51 (6.0%)	0.001	27 (12.2%)	15 (6.8%)	0.052
Laboratory parameters						
WBC (10^9^/L)	13.6 ± 6.1	10.0 ± 3.9	<0.001	13.4 ± 5.9	10.8 ± 5.2	<0.001
Hgb (g/dL)	12.5 ± 2.1	12.6 ± 2.0	0.222	12.5 ± 2.1	12.4 ± 2.1	0.749
Platelet (10^9^/L)	255.0 ± 113.5	230.2 ± 87.9	<0.001	250.7 ± 106.0	235.3 ± 92.8	0.106
Creatinine (mEq/L)	1.1 ± 0.5	1.0 ± 0.8	0.316	1.0 ± 0.4	1.0 ± 0.8	0.528
BUN (mg/dL)	22.1 ± 14.4	17.4 ± 10.4	<0.001	20.8 ± 12.4	19.1 ± 11.4	0.143
Sodium (mmol/L)	141.0 ± 5.1	140.4 ± 3.5	0.057	140.8 ± 4.8	140.7 ± 3.8	0.818
Potassium (mmol/L)	4.3 ± 0.7	4.2 ± 0.6	<0.001	4.3 ± 0.7	4.2 ± 0.6	0.006
Bilirubin (mg/dL)	0.7 ± 0.5	0.7 ± 0.7	0.917	0.7 ± 0.5	0.7 ± 0.5	0.516
Drugs, n						
Anti-platelet agents	176 (74.6%)	690 (81.3%)	0.023	165 (74.3%)	173 (77.9%)	0.373
Anti-coagulation agents						
Warfarin	51 (21.6%)	216 (25.4%)	0.227	46 (20.7%)	61 (27.5%)	0.096
NOAC	5 (2.1%)	59 (6.9%)	0.005	4 (1.8%)	10 (4.5%)	0.103
tPA or EVT	56 (23.7%)	203 (23.9%)	0.954	54 (24.3%)	62 (27.9%)	0.387
APS III	49.1 ± 24.1	34.6 ± 16.0	<0.001	45.7 ± 19.9	45.7 ± 19.5	0.988
ICU mortality, n	45 (19.1%)	42 (4.9%)	<0.001	39 (17.6%)	24 (10.8%)	0.041
ICU length of stay, day	5.5 ± 6.8	3.8±5.1	<0.001	5.2 ± 6.7	5.2 ± 8.1	0.910
In-hospital mortality, n	66 (28.0%)	73 (8.6%)	<0.001	59 (26.6%)	42 (18.9%)	0.054
Hospital length of stay, day	10.2 ± 11.1	7.0 ± 7.7	<0.001	10.0 ± 11.0	9.1 ± 10.6	0.404
Intracranial hemorrhage, n	27 (11.4%)	88 (10.4%)	0.635	27 (12.2%)	29 (13.1%)	0.775
Tracheostomy, n	17 (7.2%)	35 (4.1%)	0.050	16 (7.2%)	10 (4.5%)	0.225
PEG/PEJ tube placement, n	42 (17.8%)	94 (11.1%)	0.006	39 (17.6%)	36 (16.2%)	0.704

Propensity score matching by age, sex, Charlson comorbidity Index, acute physiology score III, the CHA2DS2-VASc score, and HAS-BLED score. APS III: acute physiology score III; BPM: beats per minute; BUN: blood urea nitrogen; CCI: Charlson comorbidity Index; COPD: chronic obstructive pulmonary disease; EVT: endovascular mechanical thrombectomy; Hgb: hemoglobin; MAP: mean arterial pressure; NOAC: novel oral anticoagulant; PVD Peripheral vascular disease; PEG: percutaneous endoscopic gastrostomy; PEJ: percutaneous endoscopic jejunostomy; tPA: tissue plasminogen activator; WBC: white blood cell. #: Testing by Fisher exact test or Wilcoxon Test.

**Table 2 jcm-11-05116-t002:** Association between outcomes and hyperglycemia among patients with ischemic stroke.

	With Hyperglycemia Versus Non-Hyperglycemia (*Reference*)			
Variables	Before PSM-Univariate		After PSM-Univariate	
Outcomes	Crude HR (95%CI)	*p* Value	Adjusted HR (95%CI)	*p* Value
ICU Mortality	2.55 (1.67−3.89)	<0.001	1.54 (0.92−2.57)	0.097
In-hospital Mortality	2.51 (1.79−3.52)	<0.001	1.33 (0.90−1.98)	0.156

Propensity score matching by age, sex, Charlson comorbidity Index, acute physiology score III. HR: hazard ratio.

## Data Availability

All data accessed and analyzed in this study are available in the article.

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
