# Peer review of "The Role of Nondiabetic Hyperglycemia in Critically Ill Patients with Acute Ischemic Stroke"

_jcm, 2022, doi:10.3390/jcm11175116_

Round 1

Reviewer 1 Report

Article “The Role of Nondiabetic Hyperglycemia in Critically Ill Patients with Acute Ischemic Stroke” studied an interesting and potentially clinically important association between nondiabetic hyperglycemia and the short-term prognosis of critically ill patients with acute ischemic stroke. The only concerns that should be addressed by the authors refer to English editing and rephrasing of some parts of the manuscript to improve overall quality and clearness. However, I would like to emphasize that mentioned concerns are not negligible and should be addressed/improved with much attention and in detail.

Abstract

In general abstract seems poorly written with grammatical/typing errors. Please, correct the errors and improve abstract in general.

Line 16: “We aims…” please correct to we aim

Lines 18-19:  “The primary outcomes were set as intensive care units (ICU) and in-hospital mortality. ear association between serum glucose levels and primary outcomes”. Please correct these typing errors, the sentence is unclear.

Lines 28-29: “Patients without diabetes with acute ischemic stroke and hyperglycemia have poor clinical characteristic compared with those without hyperglycemia; therefore, careful monitoring is required.” The sentence is unclear, please rephrase. Additionally, this is not an appropriate sentence for abstract conclusion, I would suggest replacing it with some other closing sentence/remark.

Introduction

Lines 39-40: “Although mortality rate after acute ischemic stroke (AIS) is relatively low, approximately 5%; however, the mortality rate would increase by twofold among critically ill patients with AIS[5,6].” The sentence is unclear, please rephrase. Also, the abbreviation AIS was introduced twice, please correct (the AIS was mentioned first time in Line 38).

Line 46:  “….brain ischemic cascades and further raise oxidative stress levels….”  I would suggest replacing “further raise oxidative status levels” to further compromise oxidative status.

Line 61-63: Please rephrase the sentence.

Discussion

Lines 218-219: “Increased mortality and elevated glucose levels have been hypothesized to be associated with impaired brain–blood barrier integrity…” please rephrase to have been associated.

Lines 247-249: “Consequently, regarding the maximum value of serum glucose in the first 24 h as a prognostic factor is reasonable. Furthermore, continuous monitoring to prevent patients from falling into the plateaus of our sigmoid curves should be considered.” These two sentences are unclear and are not written using academic vocabulary. Please rephrase and correct.

In general, discussion part covering lines 232-301 is very poorly written and not acceptable as a part of scientific vocabulary, it represents a simple stating of previous studies (two long, should be shorten). Please correct the whole mentioned part of Discussion, it will improve the manuscript quality very much.

Conclusion

In general, conclusion section should be improved in terms of vocabulary.

Lines 309-310: “Yet, whether there is a need of acute management for the elevated glucose level remains controversial” What is the meaning of this sentence? Please, rephrase.

Author Response

Reviewer Comments:

Reviewer #1:
Article “The Role of Nondiabetic Hyperglycemia in Critically Ill Patients with Acute Ischemic Stroke” studied an interesting and potentially clinically important association between nondiabetic hyperglycemia and the short-term prognosis of critically ill patients with acute ischemic stroke. The only concerns that should be addressed by the authors refer to English editing and rephrasing of some parts of the manuscript to improve overall quality and clearness. However, I would like to emphasize that mentioned concerns are not negligible and should be addressed/improved with much attention and in detail.

1) In general abstract seems poorly written with grammatical/typing errors. Please, correct the errors and improve abstract in general.

Response:

Thanks for providing your precious comments on our manuscript, sincerely. We would carefully revise the text according to your detailed suggestions and make the content clearer and easier to understand.

2) Line 16: “We aims…” please correct to we aim

Response:

Thanks for your comments. We have revised the description.

Revised in manuscript:

‘‘Abstract’’ section

We aim to elucidate the association between nondiabetic hyperglycemia and the short-term prognosis of critically ill patients with acute ischemic stroke.

3) Lines 18-19: “The primary outcomes were set as intensive care units (ICU) and in-hospital mortality. ear association between serum glucose levels and primary outcomes”. Please correct these typing errors, the sentence is unclear.

Response:

Thanks for your comments. We have revised the typing errors and made the sentence clearer.

Revised in manuscript:

‘‘Abstract’’ section

The primary outcomes were set as intensive care units (ICU) and in-hospital mortality. We developed a Cox proportional hazards model to determine the nonlinear association between serum glucose levels and primary outcomes.

4) Lines 28-29: “Patients without diabetes with acute ischemic stroke and hyperglycemia have poor clinical characteristic compared with those without hyperglycemia; therefore, careful monitoring is required.” The sentence is unclear, please rephrase. Additionally, this is not an appropriate sentence for abstract conclusion, I would suggest replacing it with some other closing sentence/remark.

Response:

Thanks for your comments. We have rephrased the sentence.

Revised in manuscript:

‘‘Abstract’’ section

Non-diabetic patients with acute ischemic stroke have poor clinical characteristic while encountering hyperglycemic events; therefore, careful monitoring in the acute phase is still required.

Introduction

5) Lines 39-40: “Although mortality rate after acute ischemic stroke (AIS) is relatively low, approximately 5%; however, the mortality rate would increase by twofold among critically ill patients with AIS[5,6].” The sentence is unclear, please rephrase. Also, the abbreviation AIS was introduced twice, please correct (the AIS was mentioned first time in Line 38).

Response:

Thanks for your comments. We have rephrased the sentence and corrected the abbreviation.

Revised in manuscript:

‘‘Introduction’’ section

However, although the mortality rate after AIS is relatively low in real-world experience, less than 6%; it could get a two-fold increase among the critically ill AIS patients[5,6].

6) Line 46: “….brain ischemic cascades and further raise oxidative stress levels….”  I would suggest replacing “further raise oxidative status levels” to further compromise oxidative status.

Response:

Thanks for your comments. We have revised the descriptions.

Revised in manuscript:

‘‘Introduction’’ section

Persistent hyperglycemia alters mitochondrial function and increases free radicals, which can aggravate brain ischemic cascades and further compromise oxidative status.

7) Line 61-63: Please rephrase the sentence.

Response:

Thanks for your comments. We have rephrased the sentence.

Revised in manuscript:

‘‘Introduction’’ section

We, therefore, aim to study the association between hyperglycemia and non-diabetic patients suffering from AIS and elucidate the relationship between those characteristics and subsequent clinical outcomes, hoping to provide some recommendations for physicians in daily clinical practice.

Discussion

8) Lines 218-219: “Increased mortality and elevated glucose levels have been hypothesized to be associated with impaired brain–blood barrier integrity…” please rephrase to have been associated.

Response:

Thanks for your comments. We have rephrased the sentence.

Revised in manuscript:

‘‘Discussion’’ section

Increased mortality and elevated glucose levels have been associated with impaired brain–blood barrier integrity, which allows the influx of inflammatory cells and blood solutes into the brain, leading to adverse stroke outcomes, such as neuronal death and brain edema[23,24].

9) Lines 247-249: “Consequently, regarding the maximum value of serum glucose in the first 24 h as a prognostic factor is reasonable. Furthermore, continuous monitoring to prevent patients from falling into the plateaus of our sigmoid curves should be considered.” These two sentences are unclear and are not written using academic vocabulary. Please rephrase and correct.

Response:

Thanks for your comments. We have rephrased and corrected the paragraph.

Revised in manuscript:

‘‘Discussion’’ section

Consequently, it is reasonable to regard the maximum value of serum glucose in the first 24 h after AIS as a prognostic factor. Furthermore, continuous monitoring to prevent patients’ serum glucose level from falling into the bilateral plateaus of our sigmoid curves should be kept in mind.

10) In general, discussion part covering lines 232-301 is very poorly written and not acceptable as a part of scientific vocabulary, it represents a simple stating of previous studies (two long, should be shorten). Please correct the whole mentioned part of Discussion, it will improve the manuscript quality very much.

Response:

Thanks for your comments. We have rephrased the paragraph and made the sentence more concise.

Revised in manuscript:

‘‘Discussion’’ section

Hyperglycemia is a common event after ischemic stroke[11]. A recently study has revealed the prevalence of hyperglycemia is 22% in non-diabetic AIS patients[27,28]. Similar trends were reproduced in our preliminary analysis (21.8%). According to the clinical recommendation from American Diabetes Association[29], hyperglycemia is defined as serum glucose levels of more than 140 mg/dL (7.8 mmol/L) among hospitalized patients, and the targeted range is 140–180 mg/dL (7.8–10.0 mmol/L). A similar perspective was also supported by the NICE-SUGAR trial[30]. However, for PSH, the definition varies widely with the cutoff points ranging from 110 mg/dL (6.1 mmol/L) to 180 mg/dL (10.0 mmol/L)[31-34], let alone a precise guidance. In our analysis, the regression model demonstrated sigmoid curves between the glycemic status and mortality; besides, the HR of mortality got approximately two-fold higher when glucose levels reached 140 mg/dL. Consequently, it is reasonable to regard the maximum value of serum glucose in the first 24 h after AIS as a prognostic factor. Furthermore, continuous monitoring to prevent patients’ serum glucose level from falling into the bilateral plateaus of our sigmoid curves should be kept in mind.

In a randomized clinical trial, the Stroke Hyperglycemia Insulin Network Effort (SHINE), among AIS patients with hyperglycemia, neither intensive nor standard glycemic control revealed a significant difference in functional outcomes at 3 months [35]. Meanwhile, another study has proposed that strict glycemic control using insulin improved the National Institutes of Health Stroke Scale (NIHSS) after 30 days [36]. In the UK Glucose Insulin in Stroke Trial (GIST-UK), insulin-based regimens indeed lowered the mean plasma glucose level; however, the 90-day mortality did not show a significant reduction[37]. To sum up, the clinical benefit of intense glucose level control remains ambiguous. Yet, hypoglycemia events were observed more frequently in the strictly controlled group, which is highly correlated with subsequent mortality and morbidity [38]. To some degree, our results also worked in concert with SHINE and GIST-UK. After excluding the clinical heterogeneity, hyperglycemia was not independent of the short-term prognosis in non-diabetic AIS patients. As for diabetic AIS patients, hypoglycemia events are frequent due to defective glucose counter regulation in advanced diabetes mellitus. These evidence all make physicians more hesitated in determining whether to treat PSH or not. In this study, we realized that the HR of mortality was low when the glucose level was below 140 mg/dL; moreover, the HR got mild increase when the curves approached the left side. Similar viewpoints were confirmed in the latest subgroup analysis of SHINE [38].

In a multicenter, prospective cohort study, the Korean Stroke Cohort for Functioning and Rehabilitation, Yoon et al. have highlighted that PSH also affects long-term functional outcomes [27]. Similar current was also noted in other cardiovascular diseases, such as acute coronary syndrome [39,40] and small-vessel disease, especially non-fatal lacunar stroke [41, 42], which showed long-termed protective effect and even regenerative potential with strict glycemic control. Because of the nature of the MIMIC-IV database, we chose intracranial hemorrhage, tracheostomy, and percutaneous endoscopic gastrostomy/jejunostomy tube placement as the indicators of functional outcomes. Among them, intracranial hemorrhage plays a key role, especially because it significantly affects patient’s disability and quality of life[43,44]. Even though this study failed to show significant differences after PSM, we should always consider that elevated glucose levels put patients at risk of hemorrhagic transformation, particularly those receiving reperfusion therapy.

The advantages of this study were as follows: this is a broad-scale study based on real‐world patient data and this study explored the association between hyperglycemia and survival status among patients with AIS. Nevertheless, the results should be clarified in the context of the several limitations. First, selection bias exists because the clinical information was obtained from a single institution. Second, patients with ischemic stroke are defined according to the first sequence diagnosis code in this retrospective study; however, the diagnostic accuracy remained unknown and misclassifications may result in false associations. Third, retrospective studies may have baseline differences; therefore, we adjusted as many potential confounders as possible to achieve appropriate balance using PSM. Fourth, given the limitations of the MIMIC-IV database, some important data were lacking, including the severity scale (e.g., NIHSS), the subtypes of ischemic stroke, cardiac parameters, routine biochemistry exams for stroke risk assessment (e.g., HbA1c value), the management for hyperglycemia, and the cause of mortality. No long-term follow-up data were provided; thus, we cannot get the modified Rankin Scale score at 3 months. Finally, because of the nature of observational databases, further randomized clinical trials are needed to validate our findings.

Conclusion

In general, conclusion section should be improved in terms of vocabulary.

11) Lines 309-310: “Yet, whether there is a need of acute management for the elevated glucose level remains controversial” What is the meaning of this sentence? Please, rephrase.

Response:

Thanks for your comments. We have rephrased the sentence.

Revised in manuscript:

‘‘Conclusion’’ section

Yet, it is still controversial whether there is a need of strict management for hyperglycemic events.

Reviewer 2 Report

In this retrospective single center study, performed on 1,086 subjects, the authors observed that patients without diabetes with acute ischemic stroke and hyperglycemia have poor clinical characteristics compared to those without hyperglycemia, but after matching the propensity score, the short-term outcomes (mortality and hospitalization duration) were non-significantly different.

The paper is very interesting and well written. The conclusions are supported by results. Figures are clear. The study has several limitations that have been highlighted quite well by the authors themselves.

However, this reviewer raises some issues that need to be addressed.

1- The authors in the exclusion criteria write that diabetic subjects were excluded from the study. How was the diagnosis of diabetes made (history, drug therapy?). During an acute episode such as an ischemic stroke it is known that fasting blood glucose and oral load with 75 grams of glucose are not suitable for diagnosing diabetes, but an HbA1c value, which expresses the mean blood glucose for the previous 3 months, was performed in these patients upon admission to hospital? This issue should be addressed in the methods, and possibly added to the study limitations.

2- The main limitation of the study, which could influence the results, is the lack of follow-up, even only in the medium term, which could demonstrate a possible impact of hyperglycemia on the mortality and morbidity of these subjects after discharge. In fact, it is known that hyperglycemia in non-diabetic subjects during acute events such as STEMI affects hospital mortality and post-discharge mortality and morbidity and that optimized glycemic control can improve both short- and medium-term outcomes (1- Journal of Clinical Endocrinology and Metabolism Volume 97, Issue 3, March 2012, 933-942. doi: 10.1210/jc.2011-2037.   2 - Diabetes Res Clin Pract. 2021 Aug; 178:108959. doi: 10.1016/j.diabres.2021.108959). These important issues, with the references above, need to be addressed in the discussion and added into the study limitations.

Author Response

Reviewer Comments:

Reviewer #2:
In this retrospective single center study, performed on 1,086 subjects, the authors observed that patients without diabetes with acute ischemic stroke and hyperglycemia have poor clinical characteristics compared to those without hyperglycemia, but after matching the propensity score, the short-term outcomes (mortality and hospitalization duration) were non-significantly different.

The paper is very interesting and well written. The conclusions are supported by results. Figures are clear. The study has several limitations that have been highlighted quite well by the authors themselves.

However, this reviewer raises some issues that need to be addressed.

1) The authors in the exclusion criteria write that diabetic subjects were excluded from the study. How was the diagnosis of diabetes made (history, drug therapy?). During an acute episode such as an ischemic stroke it is known that fasting blood glucose and oral load with 75 grams of glucose are not suitable for diagnosing diabetes, but an HbA1c value, which expresses the mean blood glucose for the previous 3 months, was performed in these patients upon admission to hospital? This issue should be addressed in the methods, and possibly added to the study limitations.

Response:

We are very grateful to hear from for your insightful comments on our manuscript. The diagnosis of diabetes mellitus in the database we used was based on International Classification of Diseases (ICD) codes. MIMIC-IV database is a single-center database which comprises patients’ information in critical care units, but it is not specialized for ischemic stroke patients. Therefore, there is a lack of complete record of HbA1c value. We would add this issue into our study limitations.

Revised in manuscript:

‘‘Discussion’’ section

Third, retrospective studies may have baseline differences; therefore, we adjusted as many potential confounders as possible to achieve appropriate balance using PSM. Fourth, given the limitations of the MIMIC-IV database, some important data were lacking, including the severity scale (e.g., NIHSS), the subtypes of ischemic stroke, cardiac parameters, routine biochemistry exams for stroke risk assessment (e.g., HbA1c value), the management for hyperglycemia, and the cause of mortality.

2) The main limitation of the study, which could influence the results, is the lack of follow-up, even only in the medium term, which could demonstrate a possible impact of hyperglycemia on the mortality and morbidity of these subjects after discharge. In fact, it is known that hyperglycemia in non-diabetic subjects during acute events such as STEMI affects hospital mortality and post-discharge mortality and morbidity and that optimized glycemic control can improve both short- and medium-term outcomes (1. Journal of Clinical Endocrinology and Metabolism Volume 97, Issue 3, March 2012, 933-942. doi: 10.1210/jc.2011-2037.  2. Diabetes Res Clin Pract. 2021 Aug; 178:108959. doi: 10.1016/j.diabres.2021.108959). These important issues, with the references above, need to be addressed in the discussion and added into the study limitations.

Response:

We sincerely appreciate your valuable comments and keen questions on our manuscript. Actually, no long-term follow-up data provided is the limitation of the MIMIC-IV database. In the prognosis of neurological disorders, quality-adjusted life year is usually taken in account. As for stroke, we usually view modified Rankin Scale as the target. In order to remedy this defect, we chose intracranial hemorrhage, tracheostomy, and percutaneous endoscopic gastrostomy/jejunostomy tube placement as the indicators of functional outcomes and quality of life, hoping that this makes our study more perfect.

According to the references mentioned above, strict glycemic control shows cardioprotective effect in acute coronary syndrome and even regenerative potential for myocardium in long-term follow-up. Nevertheless, in acute ischemic stroke, the secondary prevention mainly depends on the background etiology, which is based on the well-known TOAST classification[1]. Tracing back available studies, we notice that only small-vessel occlusion, which mainly contributes to lacunar stroke, mostly non-fatal, benefits from strict glycemic control [2,3]. We would reconstruct these important issues as per your recommendation and our literature review.

References:

  1. Adams HP Jr, Bendixen BH, Kappelle LJ, et al. Classification of subtype of acute ischemic stroke. Definitions for use in a multicenter clinical trial. TOAST. Trial of Org 10172 in Acute Stroke Treatment. Stroke. 1993;24(1):35-41. doi:10.1161/01.str.24.1.35.
  2. Maiorino MI, Longo M, Scappaticcio L, et al. Improvement of glycemic control and reduction of major cardiovascular events in 18 cardiovascular outcome trials: an updated meta-regression. Cardiovasc Diabetol. 2021;20(1):210. Published 2021 Oct 18. doi:10.1186/s12933-021-01401-8.
  3. Strain WD, Frenkel O, James MA, et al. Effects of Semaglutide on Stroke Subtypes in Type 2 Diabetes: Post Hoc Analysis of the Randomized SUSTAIN 6 and PIONEER 6 [published online ahead of print, 2022 May 18]. Stroke. 2022;101161STROKEAHA121037775. doi:10.1161/STROKEAHA.121.037775.

Revised in manuscript:

‘‘Discussion’’ section

In a multicenter, prospective cohort study, the Korean Stroke Cohort for Functioning and Rehabilitation, Yoon et al. have highlighted that PSH also affects long-term functional outcomes [27]. Similar current was also noted in other cardiovascular diseases, such as acute coronary syndrome [39,40] and small-vessel disease, especially non-fatal lacunar stroke [41,42], which showed long-termed protective effect and even regenerative potential with strict glycemic control. Because of the limitation of the MIMIC-IV database, we chose intracranial hemorrhage, tracheostomy, and percutaneous endoscopic gastrostomy/jejunostomy tube placement as the indicators of functional outcomes. Among them, intracranial hemorrhage plays a key role, especially because it significantly affects patient’s disability and quality of life[43,44]. Even though this study failed to show significant differences after PSM, we should always consider that elevated glucose levels put patients at risk of hemorrhagic transformation, particularly those receiving reperfusion therapy.

‘‘References’’ section

  1. Marfella, R.; Sasso, F.C.; Cacciapuoti, F.; Portoghese, M.; Rizzo, M.R.; Siniscalchi, M.; Carbonara, O.; Ferraraccio, F.; Torella, M.; Petrella, A., et al. Tight glycemic control may increase regenerative potential of myocardium during acute infarction. J Clin Endocrinol Metab 2012 ;97:933-42, doi: 10.1210/jc.2011-2037.
  2. Caturano, A.; Galiero, R.; Pafundi, P.C.; Cesaro, A.; Vetrano, E.; Palmiero, G.; Rinaldi, L.; Salvatore, T.; Marfella, R.; Sardu, C., et al. Does a strict glycemic control during acute coronary syndrome play a cardioprotective effect? Pathophysiology and clinical evidence. Diabetes Res Clin Pract 2021;178:108959, doi: 10.1016/j.diabres.2021.108959.
  3. Maiorino, M.I.; Longo, M.; Scappaticcio, L.; Bellastella, G.; Chiodini, P.; Esposito, K., Giugliano, D. Improvement of glycemic control and reduction of major cardiovascular events in 18 cardiovascular outcome trials: an updated meta-regression. Cardiovasc Diabetol 2021;20:210, doi: 10.1186/s12933-021-01401-8.
  4. Strain, W.D.; Frenkel, O.; James, M.A.; Leiter, L.A.; Rasmussen, S.; Rothwell, P.M.; Sejersten, Ripa. M.; Truelsen, T.C.; Husain, M. Effects of Semaglutide on Stroke Subtypes in Type 2 Diabetes: Post Hoc Analysis of the Randomized SUSTAIN 6 and PIONEER 6. Stroke 2022:101161STROKEAHA121037775, doi: 10.1161/STROKEAHA.121.037775.